# Deep Q-Learning with Whittle Index for Contextual Restless Bandits: Application to Email Recommender Systems

Ibtihal El Mimouni[1,2] and Konstantin Avrachenkov[1]

[1]INRIA Sophia Antipolis
[2]Smartprofile
{ibtihal.el-mimouni, k.avrachenkov}@inria.fr

## Abstract

In this paper, we introduce DQWIC, a novel algorithm that combines Deep Reinforcement Learning and Whittle index theory within the Contextual Restless Multi-Armed Bandit framework for the discounted criterion. DQWIC is designed to learn in evolving environments typical of real-world applications, such as recommender systems, where user preferences and environmental dynamics evolve over time. In particular, we apply DQWIC to the problem of optimizing email recommendations, where it tackles the dual challenges of enhancing content relevance and reducing spam messages, thereby addressing ethical concerns related to intrusive emailing. The algorithm leverages two neural networks: a Q-network for approximating action-value functions and a Whittle-network for estimating Whittle indices, both of which integrate contextual features to inform decision-making. In addition, the inclusion of context allows us to handle many heterogeneous users in a scalable way. The learning process occurs through a two time scale stochastic approximation, with the Q-network updated frequently to minimize the loss between predicted and target Q-values, and the Whittle-network updated on a slower time scale. To evaluate its effectiveness, we conducted experiments in partnership with a company specializing in digital marketing. Our results, derived from both synthetic and real-world data, show that DQWIC outperforms existing email marketing baselines.

## 1 Introduction

In sequential decision-making problems, the Multi-Armed Bandit (MAB) framework [1] has emerged as an effective paradigm for balancing the exploration-exploitation trade-off in a widespread range of applications, including advertising [2], clinical trials [3], and web content optimization [4]. The classical MAB setting assumes that the rewards associated with each arm are stationary and independent of the previous actions of the agent (decision-maker). Yet, many situations feature complex dependencies that require advanced models. One such extension is the Contextual Multi-Armed Bandit (CMAB) problem [5], where the rewards are not only dependent on the chosen arm but also on an observed context. This setting is particularly relevant in recommender systems [2, 6], where the user's preferences can be modeled as the context. While this MAB variant captures the influence of side information, it still assumes that the rewards are identically distributed over time. However, in various real-world scenarios, the underlying reward distributions can shift over time due to changes in environmental dynamics. For instance, in network routing, traffic patterns change based on user demands, network congestion, or hardware failures. These challenges prompted the study of the Restless Multi-Armed Bandit (RMAB) framework, in many fields such as communication networks [7], healthcare [8], and web crawling [9]. In this setting, the state of each arm evolves according to a Markovian transition, even when the arm is not played, making the problem harder to solve. Several approaches have been proposed to tackle the RMAB problem, including approximate solutions based on index policies [10–13], and Reinforcement Learning (RL) techniques [14, 15].

One particularly promising direction is the combination of contextual and restless bandits, coined as Contextual Restless Multi-Armed Bandits (CRMAB). To the best of our knowledge, the only works that address CRMAB are by Chen and Hou [16] and by Liang et al. [17]. Chen and Hou [16] developed an index policy algorithm using dual decomposition, and applied it to demand response decision-making in smart grids. Liang et al. [17] combined Bayesian modeling with Thompson sampling, and focused on public health applications.

In a CRMAB setting, the reward and state transition dynamics of each arm depend not only on the state of the chosen arm but also on the observed context. This framework captures the complexities of real-world applications, notably in recommender systems, where user preferences evolve over time.

In email marketing, which is the main domain of application of our work, the task involves recommending relevant content to users via emails, and adjusting to their changing behaviors. In this use case, personalization becomes essential, particularly due to the ethical concerns related to traditional approaches, such as sending bulk email campaigns.

Proceedings of the 6th Northern Lights Deep Learning Conference (NLDL), PMLR 265, 2025.

These methods generally apply a one-size-fits-all strategy, sending the same email content to large groups of users, which leads to spam [18], and negative user experiences [19]. We elaborate more on the use case of email marketing in Section A of the appendix.

To address these challenges, we propose DQWIC, a deep Q-learning algorithm that draws inspiration from Deep Q-Networks (DQN) [20] and leverages two neural networks (NN): a Q-network operating on a fast time scale, and a Whittle-network operating on a slower time scale. The Q-network approximates action-value functions (Q-values), which estimate the expected cumulative reward and guide the decision-maker toward actions that maximize the long-term rewards. The Whittle-network estimates Whittle indices, which were originally designed for restless bandit problems with resource constraints [10]. These indices represent the opportunity cost of activating an arm versus keeping it passive, providing a systematic approach to prioritize arm selection when resources are limited.

By integrating the Whittle index into a contextual restless bandit framework, we create a model that can scale to systems with a large number of users, adapt to both evolving user behaviors and context, and optimize long-term user engagement. Our approach builds upon recent works that have explored the integration of Whittle indices with deep learning [21–24].

In this paper, we extend the application of Whittle index heuristic to a contextual setting, and leverage deep reinforcement learning to handle large state spaces and context spaces. In addition, the inclusion of context allows us to handle many heterogeneous users in a scalable way. We also demonstrate the practical applicability of this approach in an email recommender system, where it shows a potential to address ethical concerns related to intrusive marketing practices.

The algorithm's utility lies in its ability to handle dynamic environments where the agent has to make sequential decisions while balancing exploration and exploitation, and incorporating both state and context information, therefore it can be applied to various other domains beyond email recommenders, such as: in healthcare to optimize personalized treatment depending on the patient's health state and context (such as age, medical history, and current symptoms); in dynamic pricing to optimize pricing strategies depending on the demand levels and context (such as competitor prices, seasonal trends, and customer behavior), etc.

The paper is structured as follows: Section 2 formalizes contextual restless bandits. Section 3 presents the application details. Section 4 describes the proposed algorithm. Section 5 outlines the experiments conducted and discusses the results.

## 2 Contextual Restless Bandits

The problem of Contextual Restless Multi-Armed Bandits (CRMAB) is as follows: let $I = \{1, 2, \ldots, N\}$ be the set of $N$ arms. Each arm is modeled as a context-augmented Markov Decision Process (MDP), where the transition probabilities depend not only on the arm's current state but also on the context, which can vary over time. Each arm $i \in I$ is characterized by: a state space $S = \{1, 2, \ldots, |S|\}$, where $s_i \in S$ represents the state that the arm $i$ can occupy at a given time step; a context space $C = \{1, 2, \ldots, |C|\}$, where $c_i \in C$ is the side information of arm $i$; and an action space defined as $A = \{0, 1\}$, where $a_i = 1$ denotes the active action of choosing arm $i$, and $a_i = 0$ denotes the passive action of not choosing it.

The probability of transitioning from state $s_i^t$ to state $s_i^{t+1}$ for arm $i$ under action $a_i^t$ and context $c_i^t$. is given by the tensor: $P(s_i^{t+1} \mid s_i^t, a_i^t, c_i^t) = P_{s_i^t, s_i^{t+1}}^{a_i^t, c_i^t}$.

The reward function $R : S \times C \times A \times S \to \mathbb{R}$ assigns a real value to each transition, reflecting the immediate gain from moving between states given a specific action and context: $r_i^t = R(s_i^t, c_i^t, a_i^t, s_i^{t+1})$.

The objective in CRMAB is to find a policy $\pi : S \times C \to A$ that maximizes the expected cumulative reward over time, by selecting which arms to activate. This is formulated as:

$$\max_{\pi} \mathbb{E}\left[\sum_{t=0}^{\infty} \sum_{i=1}^{N} \gamma^t \, r_i(s_i^t, c_i^t, a_i^t, s_i^{t+1})\right], \qquad (1)$$

where $\gamma \in (0, 1)$ is the discount factor, which balances the value of immediate versus future rewards. This optimization is subject to the constraint that, at a time step $t$, only $M$ arms can be active simultaneously:

$$\sum_{i=1}^{N} a_i^t = M, \quad \forall t \geq 0. \qquad (2)$$

## 3 CRMAB Application Details

In the context of our email recommender system application, each user $u_i \in \mathcal{U}$ corresponds to arm $i$ in the CRMAB. For each user $u_i$, the state space $S$ represents four different levels of engagement of the user which are: opening an email, clicking on a link within the email, making a purchase, or not interacting at all.

The context space $C$ contains campaign-related features such as some details about the ongoing email campaign, seasonal promotions or special discounts; and/or user-related features such as age, location, browsing history, and user segments, etc.

The action space $A$ defines the active action ($a_i = 1$) of sending a promotional email to user $u_i$, and the passive action ($a_i = 0$) of not sending an

email to the user. In a more complex setting, this action space could include multiple actions corresponding to various types of recommendations, such as sending different types of promotional emails. However, to simplify the analysis for now, we focus on a two-action framework.

The reward function $\mathcal{R}$ should be carefully designed to guide the learning agent to prioritize actions that increase engagement levels. Reward shaping [25], which incorporates domain knowledge to nudge the algorithm towards more positive actions, can be applied to further enhance the learning quality.

The goal is to maximize the expected discounted reward while ensuring that only $M < N$ users receive promotional emails at a given time step $t$. This constraint prevents overloading users with emails.

# 4 Deep Q-learning the Whittle Index with Context

## 4.1 Whittle Index

In the CRMAB setting, the Whittle index approach offers a scalable solution to tackle arm selection. This policy, initially developed for RMAB problems [10], assigns a scalar to the state of an arm to prioritize its activation. This simplifies the decision-making process by decoupling the global optimization problem into subproblems, for each arm. In order to achieve this, the Whittle index heuristic introduces a Lagrange multiplier, $\tilde{\lambda}$, to relax the condition that only $M$ arms can be activated at each time step. This relaxation transforms the original optimization problem into an unconstrained one, making it possible to handle each arm independently while still considering the overall constraint. The Lagrangian formulation of the CRMAB objective function is expressed as:

$$\max_{\pi} \mathbb{E}\left[\sum_{t=0}^{\infty}\sum_{i=1}^{N}\gamma^t\left(r_i(s_i^t,c_i^t,a_i^t,s_i^{t+1})+\tilde{\lambda}(1-a_i^t)\right)\right],$$
(3)

where $\tilde{\lambda}$ serves as a subsidy term for not activating an arm. The Whittle index for arm $i$ is defined as the smallest subsidy $\lambda_i$ that makes the agent indifferent between activating the arm ($a_i = 1$) and not activating it ($a_i = 0$). This indifference can be expressed in terms of Q-values:

$$Q_i(s_i, 1, c_i) = Q_i(s_i, 0, c_i).$$
(4)

Therefore, the Whittle index for arm $i$ is defined as:

$$\lambda_i(s_i, c_i) = \min\{\lambda : Q_i(s_i, 1, c_i) = Q_i(s_i, 0, c_i)\},$$
(5)

where $Q_i(s_i, a_i, c_i)$ is the Q-value associated with taking action $a_i \in \{0, 1\}$ in state $s_i$ under context

$c_i$ and subsidy $\lambda$. Using the Bellman equation, the Q-values for the active and passive actions in the CRMAB framework are defined as:

$$Q_i(s_i, 1, c_i) = r_i^1 + \gamma \sum_{s_i' \in \mathcal{S}} P_{s,s'}^{1,c} V_i(s_i', c_i),$$
(6)

$$Q_i(s_i, 0, c_i) = r_i^0 + \lambda + \gamma \sum_{s_i' \in \mathcal{S}} P_{s,s'}^{0,c} V_i(s_i', c_i),$$
(7)

where: $r_i^a = r_i(s_i, c_i, a_i, s_i')$ is the immediate reward and $V_i(s_i', c_i)$ is the value function representing the expected discounted reward from state $s_i'$ under context $c_i$.

By solving equation (5), we obtain the Whittle indices of the arms. The indices in the current state profile are arranged in descending order, showcasing the top M arms to be activated.

## 4.2 Algorithm

While the Whittle index provides an approach for prioritizing arms in the CRMAB setting, computing it in a tabular form becomes infeasible due to the exponential growth of state-action-context combinations, making it computationally very expensive. To solve these scalability challenges, we propose Deep Q-learning the Whittle Index with Context (DQWIC). The algorithm uses deep neural networks to approximate the Whittle indices and Q-values, by allowing to generalize across similar states and contexts. The algorithm alternates between updating a Q-network, which estimates the action-value functions (Q-values), and a Whittle network, which computes the Whittle indices, using their respective loss functions.

At each time step, the algorithm selects actions using an epsilon-greedy strategy: with probability $\epsilon$, the algorithm randomly selects $M$ arms to activate. With probability $1-\epsilon$, it selects the top $M$ arms with the highest Whittle indices, which are computed using the current state and context. After action selection, the environment transitions to the next state, and context and rewards are observed. These experiences are stored in a replay memory $\mathcal{D}$.

On a fast time scale, we update the Q-values. A mini-batch of size $B$ is sampled from $\mathcal{D}$. Each sample consists of the current state $s_t$, action $a_t$, reward $r_t$, context $c_t$, and the next state $s_t'$. The Q-network takes as input the current state $s_t$, the Whittle indices $\lambda_\delta$, and context $c_t$, and computes the predicted Q-values for all possible actions. The target network computes the target Q-values as follows:

$$Q^{\text{target}}(s_t, a_t, \lambda_\delta, c_t) = (1 - a_t)(r_0(s_t) + \lambda_\delta)$$
$$+ a_t r_1(s_t) + \gamma \max_{a \in \{0,1\}} Q_{\theta^{tg}}(s_t', a, \lambda_\delta, c_t),$$
(8)

where $(s_t, a_t, r_t, s_t', c_t)$ represent the current state, action, reward, next state, and context in the sampled batch. $\lambda_\delta$ is $\lambda_\delta(k, c_t)$, the Whittle index for

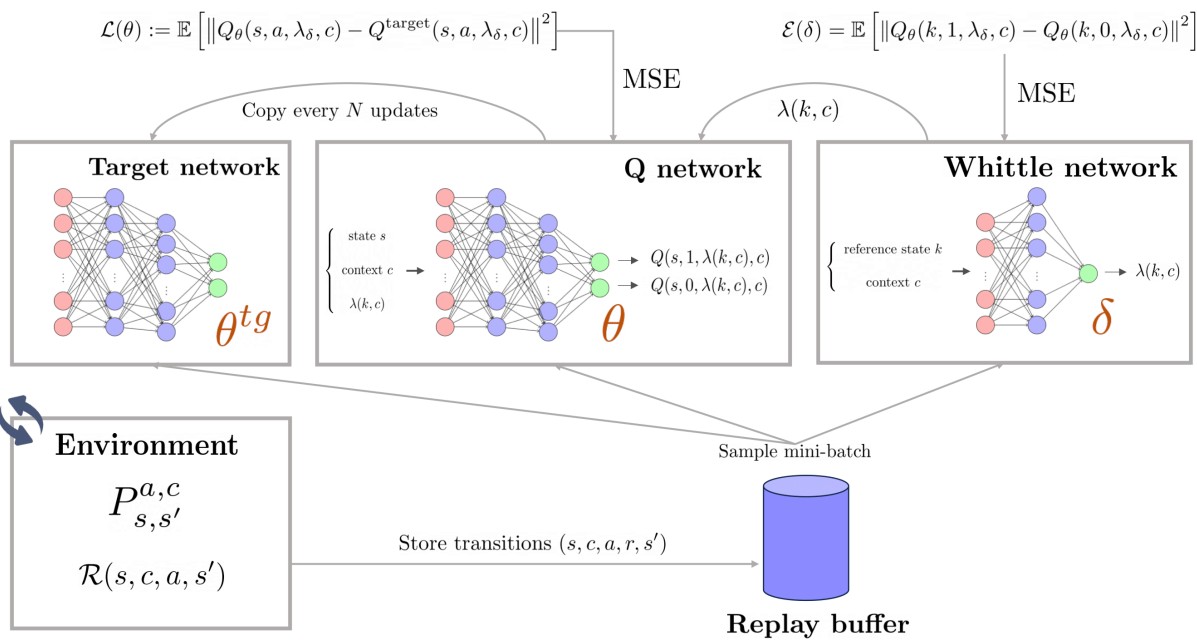

**Figure 1.** Overview of DQWIC architecture.

the reference state $k$ and context $c_t$, parameterized by $\delta$. $\max_{a\in\{0,1\}} Q_{\theta^{tg}}(s'_t, a, \lambda_\delta, c_t)$ is the maximum Q-value for the next state $s'_t$ over all possible actions, predicted by the target network parameters $\theta^{tg}$. The Q-network parameters $\theta$ are updated by minimizing the Mean Squared Error (MSE) between the predicted Q-values and the target Q-values. The loss function is defined as:

$$\mathcal{L}(\theta) := \mathbb{E}\left[\left\|Q_\theta(s,a,\lambda_\delta,c) - Q^{\text{target}}(s,a,\lambda_\delta,c)\right\|^2\right]. \tag{9}$$

The parameters $\theta$ of the Q-network are updated using backpropagation to minimize this loss. The parameters of the target network $\theta^{tg}$ are periodically synchronized with those of the Q-network $\theta$, for instance every 100 iterations, to stabilize training.

The Whittle network enables to compute the Whittle index $\lambda_\delta(k,c)$ for each state $k$ and context $c$ such that the Q-values for the active (action $a = 1$) and passive (action $a = 0$) states are equal. This indicates indifference between actions at that state. The loss function for the Whittle network is defined as the MSE between the Q-values for the active and passive actions:

$$\mathcal{E}(\delta) = \mathbb{E}\left[\left\|Q_\theta(k,1,\lambda_\delta,c) - Q_\theta(k,0,\lambda_\delta,c)\right\|^2\right], \tag{10}$$

where $Q_\theta(k,1,\lambda_\delta,c)$ is the Q-value for the active action under the current estimate of $\lambda_\delta(k,c)$, and $Q_\theta(k,0,\lambda_\delta,c)$ is the Q-value for the passive action under the current estimate of $\lambda_\delta(k,c)$.

On a slower time scale, a batch of reference states $k$ and contexts $c$ are sampled from the replay memory $\mathcal{D}$. These samples are used to compute the MSE loss function $\mathcal{E}(\delta)$ based on the difference between the Q-values for the active and passive actions. The parameters $\delta$ of the Whittle network are updated using gradient descent to minimize the loss $\mathcal{E}(\delta)$.

The algorithm pseudocode is given in Section B of the appendix.

# 5 Experiments

## 5.1 Baselines

To evaluate our proposed algorithm DQWIC, we compare it against several baseline policies commonly adopted in email marketing and restless bandit problems. Specifically, we consider the following:

- **Myopic policy:** selects to contact users who are most likely to lead to immediate conversions, such as those who have recently opened or clicked on emails;

- **Random policy:** involves sending emails to randomly selected users;

- **Round-robin policy:** sends emails evenly in a cyclic order, ensuring each user is contacted in turn, regardless of their likelihood to engage;

- **Q-index policy:** prioritizes users who are most likely to engage immediately. It is a learning-based policy that relies on a simplified greedy DQN-based approach. The Q-values are learned in the same way as in DQWIC. However, we do not compute Whittle indices. Instead, during the action selection process, we only consider

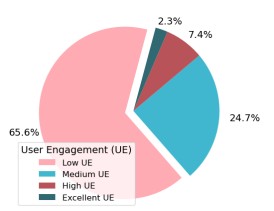

**(a)** User Engagement (UE)

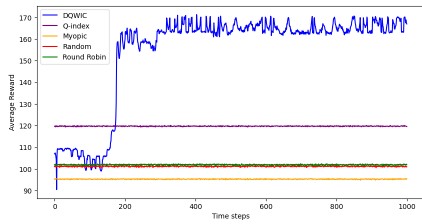

**(b)** Real-world data / N = 100, M = 10

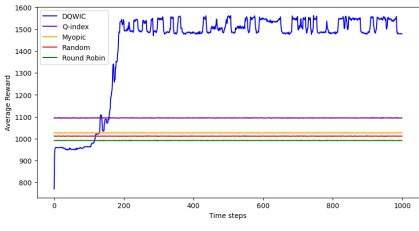

**(c)** Synthetic data / N = 1000, M = 100

**Figure 2.** Performance graphs for DQWIC algorithm vs. Q-index, myopic, round robin, and random policies. N is the total number of arms, and M is the number of selected arms.

the Q-value for action 1 ($Q_{a_1}$) for each user. Then, we rank users based on their $Q_{a_1}$ values and select the top M. In the emailing use case, action 1 means deciding to send an email. The Q-value $Q_{a_1}$ reflects the likelihood that the user will engage. The goal of this DQN-only Q-index policy is to provide a comparison with DQWIC, showing the effect of learning without the Whittle network.

We conducted experiments using both synthetic data and real-world data provided by our partner company Smartprofile [26], which is specialized in digital marketing.

## 5.2 Real-world Dataset

In a typical setup of the partner company, Smartprofile, *M* represents 10% of the potentially targeted users. This proportion was also used in our experiments. The dataset, provided by Smartprofile, was collected with user consent adhering to ethical standards, including GDPR [27] and CNIL [28] regulations. The dataset contains more than 10000 distinct users. We worked in a fully observable setup: at each time step, we can observe the state of each user in the logged anonymized dataset. Figure 2(a) illustrates the distribution of User Engagement (UE), highlighting that 65.6% of users exhibit low UE. This means that users are likely to remain Idle with low probabilities of moving to more active states like Open, Click, and Purchase. This results in data sparsity, as there are fewer interactions to analyze, making it challenging to derive meaningful insights. To address this issue and create a more balanced model of user behaviors, we developed a simulator. This also enables experiments without exposing the entire dataset, thereby ensuring more privacy. Despite starting with data of more than 10000 users, we were able to reliably construct irreducible MDPs for 100 users due to sparse data. The time-independent context we used was the location. Given the constraints of data sparsity in the original dataset, we built an additional simulator using synthetic data.

## 5.3 Synthetic Data

Aside from user location, other time-independent contexts we used include: city, age, and marital status, based of distributions from [29]. We created four user classes with transition matrices for low, medium, high, and excellent UE. Various setups reflecting different user distributions were tested, and DQWIC consistently achieved the best results. Figure 2(a) shows a setup reflecting the real-world user distribution, where low UE is most prominent. The performance in this setup exceeded that of the real dataset results because the transition probabilities, determined in collaboration with the partner company, represented more diversely user behavior in terms of opens, clicks, and purchases. This suggests that accurate user modeling through well-calibrated transition probabilities is essential for optimizing system performance.

## 5.4 Architecture of Neural Networks

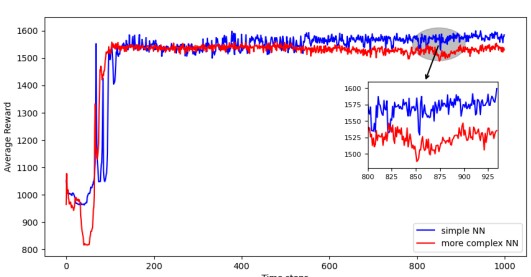

**Figure 3.** Average reward comparison of DQWIC with two different neural network architectures. N = 1000, M = 100.

In the reported experiments, we designed a relatively simple NN architecture: the Q-network featured two hidden layers with 130 and 50 neurons, respectively, while the Whittle network was constructed with a single hidden layer containing 250 neurons. For both networks, the layers are connected by ReLU activation functions.

We also tried other architectures where we increased the depth and width of the Q-network and Whittle network. For instance, a more complex NN

architecture was implemented where both networks featured four hidden layers consisting of 512, 256, 128, and 64 neurons. Additionally, batch normalization was applied after the first layer, and ReLU activation functions were used between all layers.

Figure 3 shows that the simple NN architecture outperforms the more complex NN in terms of both convergence speed and average reward. This suggests that while moderate increases in model complexity may enhance learning, excessive complexity can degrade the learning in terms of both performance and efficacy.

## 5.5 Results

Experiments, on real-world dataset and on synthetic data, were conducted over 1000 evaluation steps, representing for instance the email campaigns. Figure 2(a) illustrates the distribution of user engagement in the real-world dataset. Figure 2(b) shows the average reward comparison over time on real-world dataset, with N = 100 and M = 10. Figure 2(c) depicts the average reward comparison over time using synthetic data, with N = 1000 and M = 100. The experiments conducted, on both synthetic and real-world data, show that DQWIC demonstrates a significantly better performance in terms of average rewards exceeding Q-index, myopic, random, and round robin baselines.

## 6 Conclusion

To the best of our knowledge, we are the first to propose deep Q-learning with Whittle index for CRMAB, and to apply it to an email recommender system. One important feature of our algorithm is the usage of only three neural networks for many (potentially thousands) heterogeneous users. This is possible thanks to the addition of context to the RMAB model. Through experiments on both synthetic and real-world data, we demonstrate that DQWIC outperforms existing baselines by a significant margin. Future work will focus on incorporating multiple actions, and enhancing fairness.

## Acknowledgments

This research was conducted in collaboration between INRIA and Smartprofile, with support from the ANRT (Association Nationale de la Recherche et de la Technologie). Special thanks to our colleague Hervé Baïle from Smartprofile for his guidance and help throughout this project.

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

# A  Motivational Insights on User Modeling in Email Marketing

In modern marketing, email campaigns are a key channel for reaching out to customers. However, the traditional approach of sending bulk emails often leads to spam [18] and a negative user experience [19]. Personalized emails can significantly improve user engagement, as users are more likely to interact with content that is aligned with their preferences, behavior, and context. This makes email marketing suitable for modeling as a recommender system, we can dynamically tailor content for each user. This enables more relevant emails, which not only increases the effectiveness of the campaign but also improves user experience by delivering content they are genuinely interested in.

The email recommender system involves a trade-off between exploration (recommending to new users) and exploitation (targeting users that are already known to be engaging). The Multi-Armed Bandit (MAB) framework [1] is well-suited for this type of sequential decision-making problem, as it optimizes this balance by learning from user feedback over time.

A standard MAB model assumes that the state of each arm (in this case, each user) is static when the arm is not selected. However, in real-world scenarios (like in email marketing), user behavior evolves over time, even when no interaction occurs. For instance: a user's interests may shift due to changes in their preferences, they also might become less engaging over time if they receive too many irrelevant emails.

This dynamic nature makes the problem a restless bandit framework that allows us to model the evolving behavior of users, making it a better fit for email marketing than a standard bandit model. It captures the fact that a user's state (likelihood of engagement) can change regardless of whether an email is sent to them, which is efficient for long-term personalization.

While the restless bandit framework captures the dynamic nature of user behavior, it does not account for context, which is vital in real-world scenarios where a user's decision to engage is often influenced by contextual factors. To address this, we model

the email recommender problem as a contextual restless bandit. This approach not only considers the dynamic state transitions of each user but also incorporates the influence of contextual features.

The challenge lies in determining which users (arms) to prioritize at each time step, especially when resources (such as the number of emails that can be sent) are limited. This is where the Whittle index comes into play. The Whittle index is a well-known heuristic in restless bandit problems [10]. It simplifies the decision-making process by assigning a score to each arm, representing the value of taking an action for that arm. The arms with the highest indices are the ones that are most valuable to engage with at that moment.

Moreover, this policy is effective in balancing immediate engagement opportunities (exploitation) with long-term benefits (exploration). Some users may need immediate attention to maintain engagement, while others may benefit from receiving fewer emails. The Whittle index helps prioritize users based on their current state and context, balancing the need to engage users who are likely to respond now with the need to maintain long-term engagement.

In email marketing, with potentially thousands or even millions of users, the system needs a computationally efficient method to decide which users to target. The Whittle index reduces the complexity of selecting actions by decoupling decisions for each user, making it scalable for real-world applications. This scalability is a key strength of the Whittle index-based decomposition, allowing to scale experiments to larger datasets.

In fact, in our experiments with N = 1000 total users and M = 100 selected users per time step, DQWIC outperformed the other policies, as shown in Figure 2(c). To further validate the scalability of our method, we conducted additional experiments with a larger dataset of N = 5000 and M = 500, which corresponds closer to a real-life email campaign. Figure A.1 shows that DQWIC continues to outperform Q-index, myopic, round robin, and random policies.

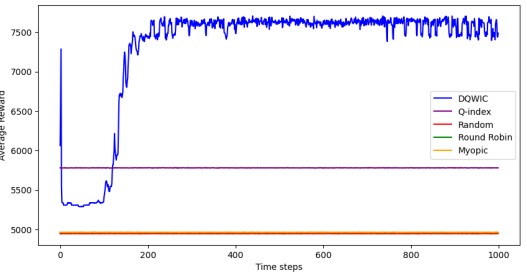

**Figure A.1.** DQWIC vs. baselines. Selection of M = 500 out of N = 5000 arms.

# B    Algorithm Pseudocode

---

**Algorithm B.1** Deep Q-learning Whittle Index with Context (DQWIC)

---

**Initialize:** Q-network parameters $\theta$, Whittle network parameters $\delta$, target network parameters $\theta^{tg}$, replay memory $\mathcal{D}$, and hyperparameters $\epsilon, \gamma$.

1: Get initial state $s$ and context $c$

2: **for** each time step $t$ **do**

3:     **if** Uniform$[0, 1] < \epsilon$ **then**
4:         Explore by selecting $M$ random arms
5:     **else**
6:         Exploit by selecting top $M$ arms with the highest Whittle indices
7:     **end if**

8:     Execute action $a$, observe context $c$, next state $s'$ and reward $r$
9:     Store transitions $(s, c, a, r, s')$ in replay memory $\mathcal{D}$

10:    **if** $|\mathcal{D}| >$ batch size **then**

11:        /* On a faster time scale */
12:        Sample mini-batch of transitions from $\mathcal{D}$
13:        Compute $Q^{\text{target}}$ (equation 8)
14:        Compute loss: $\mathcal{L}(\theta)$ (equation 9)
15:        Update Q-network parameters $\theta$

16:        /* On a slower time scale */
17:        Sample batch of $(k, c)$ from $\mathcal{D}$
18:        Compute loss: $\mathcal{E}(\delta)$ (equation 10)
19:        Update Whittle network parameters $\delta$

20:    **end if**

21:    Periodically synchronize $\theta^{tg} \leftarrow \theta$

22: **end for**

---

