# OpenReview forum: "Deep Q-Learning with Whittle Index for Contextual Restless Bandits: Application to Email Recommender Systems"
_NLDL.org/2025/Conference — NLDL 2025 Poster_

### Official Review · Reviewer_D4bP · 2024-09-23

**Confidence:** 4

**Summary:**

The authors propose to learn two networks for recommendation problems, one estimating a traditional contextual bandit and the other the Whittle index. The results are aggregated and used for email recommendation. Technically, the approach is interesting but lacks some motivation.

**Strengths:**

Interesting novel combination of MAB and Whittle index but also somewhat unmotivated (see weakness). Empirical evaluation on toy and real data.

**Weaknesses:**

The combination of MAB and Whittle index is motivated at all. The abstract should already tell us why this is a good idea. What novelty brings the second network computing the Whittle index into the game, what problem does is solve/remedy and how could we motivate the idea of bringing Whittle into this in the first place? Regarding experimentation I also see the necessity to present an ablation w/ and w/out Whittle to show that it actually has an effect and quantify how big it is, under what circumstances etc.

**Final Rebuttal Confidence:**

4

**Final Rebuttal Justification:**

I'd follow the authors arguments. It would be good to incorporate some of the rebuttal into the main text of the manuscript when preparing a possible camera-ready copy.

**Justification:**

Interesting idea that needs a better motivation and further empirical evidence.

---

> ### Author Rebuttal · Authors · 2024-10-25
>
> Dear Reviewer D4bP, thank you for your thoughtful review. We appreciate that you found our idea interesting. We address your comments in detail below, and we hope that our explanations on the motivation and baseline clarify your concerns.
>
> ---
>
> ### **1. Motivation of CRMAB and Whittle index**
>
> In order to explain our motivation of why the Whittle index has been chosen, we would like to explain the motivation behind our design choices, starting with why we model emailing as a recommender system, then why we chose to frame the problem as a contextual restless bandit.
>
> These motivations are discussed in detail in the introduction (Section 1 of the paper). Including the full explanation in the abstract would have made it too lengthy. So in the abstract, we focused on summarizing the key points. In the introduction, we explain the rationale for modeling the emailing use case as a recommender system and why the contextual restless bandit framework is particularly suited for this task. Regarding the choice of the Whittle index, we reference it in line 65 of the introduction while discussing index policies [10–13], and mention the origin of the Whittle index policy in restless multi-armed bandits (RMAB) in section 4.1 (line 202). Additionally, in the conclusion, we highlight that, to the best of our knwledge, our work is the first to propose using the Whittle index in the contextual restless bandit setting.
>
> To address your specific comments about the Whittle index motivation, we will restate in the introduction and abstract that we chose the Whittle index for its ability to efficiently prioritize actions in dynamic environments, and its proven success in balancing exploration and exploitation in restless bandit problems. By using it with contextual restless bandits, we create a model that can scale to large systems, adapt to both evolving user behaviors and context, and optimize long-term user engagement.
>
> Our motivation:
>
> In modern marketing, email campaigns are a key channel for reaching out to customers. However, the traditional approach of sending bulk emails often leads to spam and a negative user experience. Personalized email content has been shown to significantly improve user engagement, as users are more likely to interact with content that is tailored to their preferences, behavior, and context. This makes email marketing suitable for modeling as a recommender system, where we can dynamically tailor content for each user. This enables more relevant and targeted emails, which not only increases the effectiveness of the campaign but also improves user experience by delivering content they are genuinely interested in.
>
> This email recommender system involves a trade-off between exploration (recommending to new users) and exploitation (targeting users that are already known to be engaging). The multi-armed bandit (MAB) framework is well-suited for this type of sequential decision-making problem, as it optimizes this balance by learning from user feedback over time.
>
> A standard MAB model assumes that the state of each arm (in this case, each user) is static when the arm is not selected. However, in real-world scenarios (like in email marketing), user behavior evolves over time, even when no interaction occurs. For instance: a user's interests may shift due to changes in their preferences, they also might become less engaging over time if they receive too many irrelevant emails.
>
> This dynamic nature makes the problem a restless bandit framework that allows us to model the evolving behavior of users, making it a better fit for email marketing than a standard bandit model. It captures the fact that a user’s state (likelihood of engagement) can change regardless of whether an email is sent to them, which is efficient for long-term personalization.
>
> While the restless bandit framework captures the dynamic nature of user behavior, it does not account for context, which is vital in real-world scenarios where a user's decision to engage is often influenced by contextual factors. To address this, we model the email recommender problem as a contextual restless bandit. This approach not only considers the dynamic state transitions of each user but also incorporates the influence of contextual features.
>
> The challenge lies in determining which users (arms) to prioritize at each time step, especially when resources (such as the number of emails that can be sent) are limited. This is where the Whittle index comes into play.
>
> The Whittle index is a well-known heuristic in restless bandit problems. It simplifies the decision-making process by assigning a score (the Whittle index) to each arm, representing the value of taking an action for that arm. The arm with the highest index is the one that is most valuable to engage with at that moment.
>
> In email marketing, there could be thousands or even millions of users, the system needs a computationally efficient method to decide which users to target. The Whittle index reduces the complexity of selecting actions by decoupling decisions for each user. Instead of solving a large-scale optimization problem for all users simultaneously, the index allows us to rank and select the top users to engage. This makes it highly scalable for real-world applications. In fact, it has been widely used in various restless bandit applications, including communication networks, healthcare, and resource allocation, due to its ability to provide near-optimal solutions in complex dynamic environments.
>
> Moreover, this policy is effective in balancing immediate engagement opportunities (exploitation) with long-term benefits (exploration). In email marketing, some users may need immediate attention to maintain engagement, while others may benefit from receiving fewer emails. The Whittle index helps prioritize users based on their current state and context, balancing the need to engage users who are likely to respond now with the need to maintain long-term engagement.
>
> By applying the Whittle index policy to the email recommender system, in a contextual restless bandit framework, we leverage its effectiveness in managing dynamic systems, allowing for more strategic email campaigns that take into account evolving user behaviors and context influence.
>
> ---
>
> ### **2. DQN Baseline**
>
> Thank you for your feedback regarding the baseline. We have indeed presented (in Section 5.2) a DQN-based approach that operates without the Whittle index, focusing solely on Q-values to prioritize actions. It is the Q-index policy. We included this policy in order to assess the effectiveness of the Whittle index in our proposed algorithm (DQWIC) by comparing it against a learning-based approach that does not use the Whittle heuristic. Please refer to a detailed explanation of the Q-index policy in our response No. 1 to Reviewer 3bkE. Our algorithm DQWIC demonstrates a clear advantage over the Q-index baseline.

---

### Official Review · Reviewer_aNd2 · 2024-10-03
**Recommend to accept**

**Confidence:** 5

**Summary:**

This submission tries to study Deep Q-Learning with the Whittle Index for Contextual Restless Bandits problem with Application to Email Recommender Systems so as to propose DQWIC, where you aim to enhance content relevance and reduce spam messages, and you employ Q-network featured two hidden layers with 130 and 50 neurons

**Strengths:**

1. this draft is easy to follow
2. your core idea of combining deep Q-learning with the Whittle index into CRMAB is reasonable, which shares the same spirit of clustering
3. this manuscript has the potential to work on large-scale data, especially decentralised scenarios

**Weaknesses:**

1. your baselines in the experiments are a bit weak
2. the data scale adopted in section 5.3 is not massive that you're encouraged to significantly enlarge it
3. related state-of-the-art you may want to compare: Fast Distributed Bandits for Online Recommendation Systems, The Art of Clustering Bandits

**Final Rebuttal Confidence:**

5

**Final Rebuttal Justification:**

I've read the rebuttal and again, would love to recommend towards acceptance, and assume that the authors will polish based on all comments and suggestions.

**Justification:**

rest comments
-try to add more experimental results would be helpful to improve this work
-on the other hand, a theoretical analysis is also encouraged to provide to make this work more solid
Overall, it's enjoyable to have a read for this intriguing work, though there are some jobs that need to be done to better shape your merits, in short, it would be a pleasure to recommend towards acceptance.

---

> ### Author Rebuttal · Authors · 2024-10-25
>
> Dear Reviewer aNd2, thank you for your insightful comments and suggestions. We are pleased that you found our paper enjoyable to read. We address some of your comments in the following responses.
>
> ---
>
> ### **1. Choice of baselines**
>
> We understand your remark regarding the baselines. The current baselines were chosen because they are currently used by our partner company, and are commonly used by email markerters. We wanted to demonstrate improvements over existing industry practices. In addition, in order to compare with a learning-based approach, we also included the Q-index policy which is a DQN-only method that does not use the whittle network. Please refer to a detailed explanation of the Q-index policy in our response No. 1 to Reviewer 3bkE.
>
> Given that contextual restless bandits is a very recent area of research, we plan to investigate and implement additional policies in future work as they become available.
>
> We also thank you for your suggestion to explore "Fast Distributed Bandits for Online Recommendation Systems, The Art of Clustering Bandits". In fact, we had already been considering integrating clustering techniques into our approach to contextual restless bandits. This represents a potential direction for future research, as it requires substantial additional research beyond the scope of the current paper.
>
> ---
>
> ### **2. Experimental results**
>
> We agree that including more experimental results would strengthen the evaluation. While we reported on experiments with 1,000 users for the synthetic data, we also conducted experiments with 10,000 users. We will include these results in the appendix of the camera-ready paper. Furthermore, we can easily scale our experiments to even larger datasets, as the scalability is the strength of the Whittle index based decomposition.

---

### Official Review · Reviewer_aM1w · 2024-10-09

**Confidence:** 4

**Summary:**

This paper proposes a new, deep-learning based, algorithm for contextual restless bandits. The contextual restless bandit is a model which combines the dependence on exogenous features ('context') with (potentially action dependent) changing action states, and has been studied in two recent papers (Chen et al (2024) and Liang et al. (2024)). The proposed algorithm is the novel aspect of this paper, which combines two neural networks: one to learn relationships between context and rewards and one to learn Whittle indices (popular tools in tackling restless bandit problems).

A problem in email marketing serves as a motivation for the problem and algorithm, and data from this problem serve as a basis for empirical work where the algorithm is found to learn a good strategy and outperform simple benchmarks.

**Strengths:**

The paper is generally clear and correct. The challenges of context and non-stationarity are well explained and the proposed algorithm is a sensible candidate to handle these two aspects. Further to this the algorithm appears to work well on the problem, and the methodology is described in a level of detail sufficient to allow reproduction of the methods. I think there would be interest in the problem and algorithm from the NLDL community and beyond.

**Weaknesses:**

While I found the method to be appropriate to the problem posed, I did feel that the relationship between this problem and other better studied problems could be better explained.

Firstly, I struggled to find a clear explanation of how this problem differs from other contextual MDP settings. Here each arm is modelled as its own MDP and the problem is considered as a restless bandit. But if the combination of arm states was treated as the state in a wider contextual MDP (potentially partially observed) would there not be some equivalence to the current setting meaning a wider range of algorithms may be applicable?

Second, while there is little work on contextual *restless* bandits, there does seem to be a line of work on non-stationary contextual bandits that is of relevance. See e.g. Luo et al. (2018), Wu et al. (2018), Russac et al. (2019). While Whittle index based approaches may not be directly applicable here, and some of these may e.g. operate with changing arm contexts or common regression parameters in a way your model does not, it does not seem to be an entirely disconnected literature.

The other issue I found was that the experiments appear to only compare to non-adaptive approaches (though I couldn't tell whether the Q-value approach was based on a static Q-table or is adaptive but just fails to learn anything at all). This seemed a surprising choice, and I wondered how the approach would compare to either other methods from the literature, be those earlier contextual restless MAB algorithms, or choices from the non-stationary contextual bandits literature I have indicated.


Ultimately I have three questions for the authors which would influence my overall recommendation:
1. How does the problem differ from a contextual MDP? If it does, what are the situations where modelling it this way introduces a benefit and how can a practitioner be sure they are facing such a situation?
2. How do your methods and the problem relate to the literature on non-stationary contextual bandits?
3. Are all of your baseline methods non-adaptive? Is it possible to introduce an adaptive approach or explain why approaches previously existing in the literature were not suitable for comparison here?

Luo, Wei, Agrawal, Langford (2018) Efficient Contextual Bandits in Non-stationary worlds. Conference on Learning Theory
Wu, Iyer, Wang (2018) Learning Contextual Bandits in a Non-stationary Environment. SIGIR '18
Russac, Veranda, Cappe (2019) Weighted Linear Bandits for Non-stationary Environments. NeurIPS 2019.

**Final Rebuttal Confidence:**

4

**Final Rebuttal Justification:**

On balance, I feel the paper is appropriate for acceptance, but I would echo the comment of the other reviewer that aspects of the rebuttal should be (substantively) added to the paper.

**Justification:**

While the paper is clear and correct, I have outstanding concerns about its relationship to the existing literature. I have asked questions regarding this and hope that a convincing response, which either highlights a misunderstanding on my part or promises how (specifically) the paper could be updated to reflect its connections to the literature and improve limitations of the experiments could improve my rating.

---

> ### Author Rebuttal · Authors · 2024-10-25
>
> Dear Reviewer aM1w, thank you for your detailed review. We are glad that you found the paper clear and correct, and we appreciate your recognition of the relevance and interest in the problem we address.
>
> We have carefully considered your comments and provided our responses below. We hope that our explanations will answer your questions.
>
> ---
>
> ### **1. Contextual MDP**
>
> Thank you for your insightful question about the relationship between Contextual Restless Multi-Armed Bandits (CRMAB) and Contextual MDPs (CMDP). Your suggestion involves exploring whether the CRMAB problem could be reframed as a CMDP by treating the joint state of all arms (the Cartesian product of all individual arm states) as a single combined state of the environment. This would mean modeling the entire system of multiple arms (users) as a single CMDP rather than treating each arm independently.
>
> Our CRMAB model could indeed be reframed as a CMDP by considering the joint state of all users (arms) as a combined state vector. However, this approach presents significant challenges in terms of scalability and computational complexity. Below, we explain the main distinctions between CRMAB and CMDP and outline why CRMAB offers structural advantages.
>
> 1. Independent arms vs. single Markov process:
>     - In **CRMAB**, the environment consists of several independent processes, each represented as an arm. Each arm (i.e. user) is modeled as a CMDP. Each user has their own state, and this state changes over time even if we do not take action (restless dynamics).
>     - In **CMDP**, the agent interacts with a single Markov process. There is no notion of arms. We define the state space as the combination of all users' engagement states. For example, if we have 1,000 users, each of whom can be in one of 4 engagement states, the global state would be a vector like $ (s_1, s_2, ..., s_{1000}) $, where $s_i$ is the engagement state of user $i$.
>
> 2. Resource constraints:
>     - **CRMAB** naturally incorporates the idea of resource constraints by limiting how many users (arms) can be selected at each time step (e.g., we can only select M out of N users meaning we can only send M emails at a time).
>     - In **CMDP**, an explicit resource constraint is not typically modeled. While we could potentially incorporate constraints into a CMDP, doing so would further complicate the model, making it less efficient than a CRMAB approach.
>
> 3. Action policy and state space complexity:
>     - In our **CRMAB** model, we calculate a Whittle index for each user, which tells us which users would benefit the most from receiving an email at the current time step based on their state and context. The Whittle indices help us rank all users. Based on the ranking, we select the top M users to send emails to. The users with the highest Whittle indices are the ones that the model predicts will provide the most reward if they are engaged with (by sending them an email, i.e action = 1).
>     - In **CMDP**, we would need to define an action policy that decides, for the global state $ (s_1, s_2, ..., s_{1000}) $, which subset of M users should be activated (emailed). The decision would depend on the joint state of all users, rather than on individual users' states as in CRMAB. Since the state now represents the combined state of all users, the state space becomes exponentially large (curse of dimensionality). If each user has 4 possible engagement states, and there are 1,000 users, the total number of possible states for the system is $ 4^{1000} $. This is an extremely large number (more than the number of atoms in the universe) and computationally infeasible because it requires evaluating the global state and its potential changes due to the selected actions.
>
> Therefore, a practitioner should consider using CRMAB when:
>
> - Multiple independent processes (arms) need to be managed simultaneously.
> - Only a limited number of arms can be selected at each step, requiring the prioritization of actions. For example, we may only be able to send emails to a limited number of users at a time, so we need to prioritize whom to target.
> - The state of the processes (arms) continues to evolve regardless of whether the agent takes action on them. This applies in scenarios where not interacting with a process can lead to its state deteriorating or improving over time (e.g., patient health in healthcare applications, user engagement in a recommender system, etc.).
>
> The contextual restless bandit setting's structure allows for index policies (like Whittle's index) tailored for problems where each arm has separate dynamics. Index policies simplify decision-making by assigning a priority index to each arm, indicating which ones are most beneficial to activate given a resource constraint (e.g., a limited number of arms to pull). This is not straightforwardly replicable with a standard CMDP framework, where we would otherwise face the full complexity of tracking and optimizing across a large joint state space.
> The CRMAB formulation allows algorithms to exploit the decomposability across arms. This provides a structural advantage, as it allows each arm to be treated separately.
>
> ---
>
> ### **2. Contextual restless bandits vs. Contextual non-stationary bandits**
>
> Contextual non-stationary bandits and contextual restless bandits are both extensions of the multi-armed bandit problem that deal with changing environments and contextual information. However, they have some key differences.
>
> In contextual non-stationary bandits, the reward distributions of arms change over time, and the contextual information helps inform arm selection. The focus is on adapting quickly to arbitrary changes in the reward distributions.
>
> On the other hand, contextual restless bandits model a more structured form of non-stationarity. In these problems, each arm has an internal state that evolves over time according to a Markov process, regardless of whether the arm is selected or not. This "restless" nature means that arms are constantly changing, even when not interacted with. The contextual information in this case influences arm selection based on these evolving states.
>
> In addition, a key feature of contextual restless bandits is the inclusion of explicit resource constraints. Typically, these problems involve selecting M out of N arms, making them well-suited for practical applications where decisions are constrained by real-world limits such as budget or capacity. This resource allocation aspect is often not present in non-stationary contextual bandits, which typically do not incorporate hard constraints on the number of arms that can be selected simultaneously.
>
> While both types of bandits deal with changing environments and contextual information, contextual restless bandits provide a more structured approach to modeling non-stationarity through Markov chains. This makes them a distinct class of problems with different solution approaches and theoretical properties compared to contextual non-stationary bandits. The contextual restless bandit problem is more complex and is often solved using approximate methods, as finding optimal solutions is PSPACE-hard.
>
> ---
>
> ### **3. Adaptive baseline**
>
> Thank you for your feedback regarding the baselines used in our experiments.
>
> First, the Q-index baseline is indeed a learning approach. It is based on a DQN framework, where the Q-values are updated over time based on observed rewards and the system dynamically adjusts its decisions. Unlike DQWIC, the Q-index policy omits the Whittle network and uses only the Q-network to approximate Q-values for decision-making. Please refer to a detailed explanation of the Q-index policy in our response No. 1 to Reviewer 3bkE.
>
> Regarding the other baselines, they are non-adaptive. For instance, the myopic and random policies are commonly used to compare algorithms in restless bandits to establish simple benchmarks. On the other hand, we chose to compare against these non-adaptive baselines because they are currently used by our partner company and are often adopted by marketers in email marketing. The objective is to demonstrate the practical improvements of our proposed method over existing industry standards in emailing.
>
> However, we understand the importance of comparing against more adaptive methods from the literature. We did consider several algorithms from the literature, but many existing approaches were not directly applicable to our specific problem setting, which involves both context and restless dynamics. For instance, algorithms from non-stationary contextual bandits often focus on context but do not account for the restless nature of the arms.

---

### Official Review · Reviewer_3bkE · 2024-10-09

**Confidence:** 3

**Summary:**

This paper tackles the contextual restless mutli-armed bandits system problem, and proposes to approach it with a solution inspired by deep Q-learning.

The paper uses the Whittle index, that allows to re-write the CRMAB problem without constraints, by relaxing the "number of arms" constraint with a Lagrangian. Then, they use a modified DQN algorithm with an additional network. Their Q-network is learnt just like in DQn, but with an extra dependency on a parameter. This parameter is an estimation of a whitlle index, computed by the second network.

Then, the algorithm is applied to email recommender systems, on both real and synthetic data, showing a cleat improvement over rule-based baselines.

**Strengths:**

The algorithm is clearly presented and evaluated. There is no theoretical evaluation of the method (and I don,t think it is needed) but the methodology makes sense overall.

**Weaknesses:**

**Baseline**

I think one thing missing from the paper is a comparison to a baseline that would show the importance and effectiveness of the whittle network. For example, what happens if one gets rid of it completely and simply run DQN on the problem ? I think this baseline (or a similar idea) should be presented along the results.

**Scaling**

The author mention : "We also tried other architectures where we increased the complexity [...]. We noticed that a moderate increase in network capacity, can enhance performance, while excessive complexity, can degrade the learning [...] ". Scaling neural networks is usually a tricky subject in deep RL, and thus I think this experimental results would be very valuable added in the paper or appendix.

**Few comments**

- eq 4 is not very clear, it looks like all Q-values should be the same, should the Q depend on \lambda here ?
- Q-values and values are not defined properly in the paper
- the concept of whittle index is used in the introduction, but is only explained later. For the deep Rl audience, it could be useful to shortly describe what it is used for in the introduction.

**Justification:**

Paper is overall sound and clear, with no major errors or concerns.

A main improvement direction is a more relevant baseline, that would showcase the empirical role of the whittle network.

---

> ### Author Rebuttal · Authors · 2024-10-25
>
> Dear Reviewer 3bkE, thank you for your detailed and constructive feedback. We are glad that you found our paper sound and clear. We have addressed your comments in the responses below.
>
> ---
>
> ### **1. DQN baseline**
>
> Thank you for your comment regarding adding a DQN baseline. We would like to clarify that the Q-index policy (discussed in Section 5.2) addresses your question, as it applies a DQN-based method without the Whittle network. However, before explaining the Q-index baseline in more detail, we will first discuss why running a standard DQN on this problem is not feasible.
>
> In our paper, we model the email marketing problem as a Contextual Restless Multi-Armed Bandit (CRMAB), where each arm represents a user. For each user, we must decide whether to send or not send an email, with the goal of maximizing engagement over time.
>
> If we were to apply standard DQN to this problem, we would need to learn and evaluate Q-values for both possible actions (send or not send an email) for each arm (user):
>
> - $ Q_0 $: The expected reward for action = 0 (not sending an email).
> - $ Q_1 $: The expected reward for action = 1 (sending an email).
>
> Suppose that we have 1000 arms, for each arm we need to decide whether to activate it (send an email) or leave it passive (not send an email). The challenge arises from the fact that, for each arm, there are two possible actions, and the total number of possible combinations of actions across all arms is $ 2^{1000} $. This number is incredibly large; in fact, larger than the number of atoms in the universe (around $2^{265}$). It is computationally infeasible to evaluate or even store that many possibilities. Because of this, it is impossible to use standard DQN alone in this CRMAB setting.
>
> The Whittle index heuristic provides an efficient way to address this complexity. Rather than searching through all combinations of actions, the Whittle index policy allows us to compute a score (Whittle index) for each arm, which reflects how valuable it is to activate an arm (to send an email to a user) at the current time step. Then, based on the computed Whittle indices, we select the top $ M $ arms to activate (users to send emails to).
>
> This approach transforms the large-scale decision-making problem into a tractable one by decoupling the decisions for each arm, thereby making it computationally feasible to handle large numbers of arms.
>
> In addition to comparing DQWIC (our proposed algorithm) to traditional baselines like myopic and random policies (commonly used in restless bandit problems and in email marketing application in particular), we also included a Q-index baseline. The Q-index baseline omits the Whittle network and relies purely on a simplified, greedy DQN-based approach. In the Q-index policy, we learn Q-values in the same way as in DQWIC. However, we do not compute Whittle indices. Instead, during the action selection phase, we only consider the Q-value for action 1 ($ Q_1 $). After computing $ Q_1 $ for each arm, we rank arms based on their $Q_1$ values and select the top M arms.
> In the emailing use case, action 1 means deciding to send an email. The Q-value $Q_1$ reflects the likelihood that the user will engage. By focusing on users with high $Q_1$
> values, the Q-index policy prioritizes the users who are most likely to engage immediately. This makes the Q-index policy a greedy approach, as it maximizes short-term gains.
>
> The goal of using the DQN-only Q-index policy is to provide a comparison with DQWIC, showing the effect of learning without the Whittle network.
>
> For the camera-ready version, we will detail the definition of the Q-index baseline to make it clearer that it represents a DQN-based approach without the Whittle network component.
>
> ---
>
> ### **2. Scaling Neural Networks**
>
> Thank you for your suggestion regarding providing some results about scaling Neural Networks (NN).  We agree that this would complement our discussion in Section 5.1. In the revised version, we will include, in the appendix, additional details and experiments showing how different NN architectures affect the performance of our algorithm.
>
> ---
>
> ### **3. Equation 4**
>
> Thank you for your question raised on equation 4 about how the Q-values relate to $\lambda$.
> The Whittle index is the minimum value at which the decision to activate or not activate an arm $i$ becomes indifferent:
> $$\lambda_i : Q_i(s_i, 1, c_i) - Q_i(s_i, 0, c_i) = 0$$
> Thus, the Q-values expressed in equation 4 indeed depend on $\lambda$. In the camera-ready version, to clarify this further, we will either adjust the notation or explicitly state this definition. For arm $i$, given a state $s_i$ and context $c_i$, the Whittle index is defined as:
> $$\lambda_i(s_i, c_i) = \min \\{\lambda : Q_i(s_i, 1, c_i) = Q_i(s_i, 0, c_i) \\}$$
>
> This definition states that the Whittle index $\lambda_i(s_i, c_i)$ is the minimum subsidy that makes $Q_i(s_i, 0, c_i) $ and $ Q_i(s_i, 1, c_i)$ equal.
>
> ---
>
> ### **4. Definitions of Q-values and $\lambda$ values**
>
> Thank you for pointing out the need for properly defining the Q-values and $\lambda$ values in the introduction. Indeed, understanding these terms is essential for a better grasp of our proposed CRMAB model and algorithm. To address this, when discussing our algorithm DQWIC in the introduction (around line 94), we will provide a definition of both Q-values and the Whittle index.

---

### Meta-Review · Area_Chair_H5f5 · 2024-10-31

**Recommendation:** Accept (Poster)
**Confidence:** 5

**Metareview:**

The paper presents a DQN-based method combined with the Whittle index to address the CRMAB problem. It is well-written, and the overall approach appears to be sound and novel. However, there is room for improvement in positioning the proposed approach within the literature, providing a stronger motivation for modeling the problem this way, incorporating some more background on Whittle index, and clarifying the evaluation, as suggested by the reviewers. With the expectation that these will be updated in the camera-ready version, I recommend accepting the paper.

**Suggested Changes To The Recommendation:**

1: I agree that the recommendation could be moved down

---

### Decision · Program_Chairs · 2024-11-06

**Decision:**

Accept (Poster)

**Comment:**

We recommend a poster presentation given the AC and reviewers recommendations.